# Determinants of Out-of-Pocket Health Spending in Households in Peru in the Times of the Pandemic (COVID-19)

**DOI:** 10.3390/ijerph20186759

**Published:** 2023-09-14

**Authors:** Julio Cesar Quispe Mamani, Balbina Esperanza Cutipa Quilca, Rolando Cáceres Quenta, Nelly Beatriz Quispe Maquera, Betsy Quispe Quispe, Adderly Mamani Flores, Duverly Joao Incacutipa Limachi, Angela Rosario Esteves Villanueva, Vicente Málaga Apaza, Olimpia Tintaya Choquehuanca

**Affiliations:** 1Faculty of Economic Engineering, National University of Altiplano, Floral Avenue 1153, Puno 21001, Peru; 2Faculty of Accounting and Administrative Sciences, National University of Altiplano, Floral Avenue 1153, Puno 21001, Peru; becutipa@unap.edu.pe; 3Faculty of Educational Sciences, National University of Altiplano, Floral Avenue 1153, Puno 21001, Peru; rcaceres@unap.edu.pe; 4Faculty of Health Sciences, Professional School of Dentistry, National University of Altiplano, Floral Avenue 1153, Puno 21001, Peru; nbquispe@unap.edu.pe (N.B.Q.M.); betsyquispe@unap.edu.pe (B.Q.Q.); 5Faculty of Social Sciences, National University of Altiplano, Floral Avenue 1153, Puno 21001, Peru; adderlymamani@unap.edu.pe (A.M.F.); djincacutipa@unap.edu.pe (D.J.I.L.); otintaya@unap.edu.pe (O.T.C.); 6Faculty of Nursing, National University of Altiplano, Floral Avenue 1153, Puno 21001, Peru; aresteves@unap.edu.pe; 7Faculty of Chemical Engineering, National University of Altiplano, Floral Avenue 1153, Puno 21001, Peru; vmalagaa@unjbg.edu.pe

**Keywords:** comprehensive health insurance, out-of-pocket expenses, older adult, illness, socioeconomic level, COVID-19

## Abstract

In 2021, the expenses paid by households worldwide due to COVID-19 showed an increasing behavior and directly affected economic income since they were part of unforeseen expenses among households and became a factor that contributed to the increase in the levels of poverty mainly in households that were not part of the health system. The objective of this research was to establish the main determinants of out-of-pocket spending on health in Peruvian households in the times of the pandemic. A quantitative approach, of a nonexperimental type, with a descriptive and correlational methodological design was considered. The database of the National Household Survey of the National Institute of Statistics and Informatics for 2021 was used as a source of information, applying the binomial logit econometric model. Out-of-pocket expenses during the pandemic compared to normal periods were shared by the members of the households. Since they were part of unforeseen expenses, these expenses mainly impacted the heads of the households and strongly affected household budgets. For this reason, the type of insurance, the suffering of household members from a disease, the results of tests for COVID-19, the expenditure on individual health, the existence of permanent limitations to any member of the household, the presence of an older adult in the household, and the marital status of the head of the household determined and positively influenced out-of-pocket spending in households in Peru with 36.85, 8.48, 6.50, 0.0065, 23.73, 16.79, and 2.44 percentage units. However, the existence of a drinking water service in the household, educational level, and the area of residence determined and negatively influenced out-of-pocket spending in households in Peru with 4.81, 6.75, and 19.26 percentage units, respectively. The type of insurance, the suffering of an individual from a disease, the results of COVID-19 tests, health spending, the existence of permanent limitations, the presence of an older adult in the household, and the marital status of the head of the household positively determined out-of-pocket spending in households in Peru, while the existence of a potable water service, educational level, and the area of residence determined out-of-pocket expenses in a negative or indirect way.

## 1. Introduction

Relating the health sector to the economy is important in these times of global change. This has become even more apparent since the world faced a health crisis due to COVID-19, which greatly affected the economies of families and countries, directly affecting family incomes. The health problems faced by households translate into expected scenarios, where some families allocate part of their economic income to health spending. However, spending can also occur in unforeseen scenarios, and it is in these cases where out-of-pocket spending is greatly affected, as health expenses, mainly due to unforeseen phenomena, cannot be covered [1,2,3,4,5,6,7,8,9].

In this sense, at the world level, many countries have experienced changes in their demographics mainly due to the progressive growth of the school-aged and older adult populations, which has categorically affected the health sector since the demand for health services has increased. Health services mainly cover the treatment of diseases such as respiratory and cardiac conditions and diabetes; however, these services became even more complicated in the times of the pandemic, since the above age groups were the most affected and higher costs were generated in the health sector [10,11,12,13,14,15,16,17,18].

The complications that COVID-19 generated in the health sector across the world exposed the conditions in which health centers, hospitals, and health clinics operated; in developing countries, these centers were not prepared to face such health crisis conditions and found themselves in conditions of inefficiency and injustice. Many members of households could not guarantee the financing of unforeseen expenses when contracting a disease; due to the accruing expenses, they were not able to efficiently pay for medicines or hospital services. All this was due to the existence of a scenario of uncertainty due to their health condition and, in some cases, their loss of health, since despite the fact that some members of households (heads of a household or members of a household) have Comprehensive Health Insurance (SIS) or Social Health Insurance (ESSALUD), out-of-pocket spending exposes household members to the risk of being unable to afford unforeseen expenses [7,13,19,20,21].

In addition, currently, one of the most recurring problems is the lack of financing for health needs: according to the WHO, in the last year, there has been an increase in global health spending, with annual increases of 6% in countries with low and/or middle incomes and 4% in countries with high incomes [3,21,22,23,24].

Some international precedents show that, in health systems, the financing structure plays an important role in achieving universal health coverage; public spending in the health sector is insufficient and inadequately allocated in the case of Latin America. When there is a dependency on private spending on health services, health coverage is limited by the availability of financing that improves the solidarity of the health system [25]. In addition, the distribution of direct payments among the population, as long as the distribution is analyzed among the people who made the payments, is closer to the prices paid and is indicative of how a person is affected by their need for care; this research result indicates that financial protection from the national health system is weak and the out-of-pocket financing model is inequitable [26]. For example, in Mexico, the health system has been characterized as a highly fragmented system, where a high percentage of its financing—around half—is obtained through household out-of-pocket expenses. This has generated a high level of catastrophic spending, with impoverishment in some cases, and deepening poverty in others [13,14,27].

In addition, out-of-pocket spending has increased thanks to health conditions, being higher during the later periods of life. In the United States, out-of-pocket spending totals USD 5211 in the last year of life of older adults, since, in this country, out-of-pocket spending increases in relation to age and health insurance status [28]. This is corroborated by [21,29], who established that ESSALUD private spending increases as income increases in absolute terms, and out-of-pocket spending increases as a proportion of income as the socioeconomic level of the household decreases and insurance coverage decreases. The unequal distribution of education determines the use of health services, which, at the same time, can become generators of poverty. In Panama, access policies and financing of drug spending should be the focus, in addition to reducing the social gap in public health in the future [30]. Finally, in Mexico, when older adults benefit from public insurance, they also incur out-of-pocket expenses, which affects their economy, considering that most of them have low incomes. Regarding medicines, the SP only covers 34.9%. Here, it can be seen that the medicine category is essential in health, both in young people and older adults, regardless of whether they are inhabitants of urban or rural areas [14,31,32].

Peru has achieved economic expansion in recent years, allowing it to have suitable resources to be able to promote various reforms towards the social sectors, but it is also one of the countries with the lowest health spending in Latin America; therefore, health services for people are carried out through the Comprehensive Health System (SIS) and Social Health Security (ESSALUD), where by 2021, according to data from the National Institute of Statistics and Informatics (INEI), the largest number of people were insured through SIS, representing more than 60%, while an average of 37% were insured through ESSALUD, whose purpose is to guarantee and ensure protection and the right to health services to all residents of the country [33,34,35,36].

According to the indicators of the services provided by SIS and ESSALUD, they manage to provide effective financial and benefit coverage, since, in 2021, the Ministry of Health of Peru reported that the out-of-pocket spending of people who are affiliated with SIS increased from 18% in 2012 to 26% in 2019; in addition, the allocation of out-of-pocket spending oriented towards the acquisition of medicines increased to 35% in 2019, which demonstrates the existence of a gap in access to health services, which is conditioned by the payment capacity of clients or users, generating financial problems in households [33,34,35,36].

In the times of the pandemic, out-of-pocket spending represented close to 90% of household spending on health. Medicines were the fundamental out-of-pocket expenses, representing between 43 and 47% of households. One study came to the conclusion that consultation with public establishments increases the demand from the private sector, since the public sector does not have the necessary instruments for the care of patients who use its services. This concerns the provision of technology, including the supply and sale policies of pharmacies of the National Ministry of Health and ESSALUD [33]. The WHO indicates that out-of-pocket spending should not exceed between 30% and 40% of total health spending. When a country is based more on direct payments, it will be more difficult to achieve universal access to quality health services [33,34,35,36].

Out-of-pocket spending on health in households in Peru is a significant source of financing, since the negative impact on income and people’s health generates inequality in access. Peru implements public insurance that focuses specifically on the poor and vulnerable population. It is affirmed that despite this insurance, there is a high level of out-of-pocket spending, which translates to a financial risk [37].

Considering the existing information from the ENAHO database for 2020, household spending on health services exceeded PEN 10 billion, but this amount was below what was reported in 2019. In said period, on average, households managed to spend the amount of PEN 1291 in out-of-pocket expenses for medical consultations and the purchase of medicines, and in 2020, these expenses dropped to PEN 1136, decreasing by 12%. However, this behavior was not significant, nor was it uniform, given that the group of families that are part of the lowest income quintile showed an increase in out-of-pocket health spending between said periods, increasing from PEN 311 to PEN 426, evidencing an increase of 37% [33].

Some national precedents show that one of the main challenges in Peru is being able to improve the equity of health services for the population. Therefore, the World Health Organization (WHO) advises an increase in the number of studies that are carried out on health systems, since the proper functioning of health systems is related to social and economic development [38]. Findings from the health sector show that the attention given to SIS contributes to cushioning the inequity in health and the percentage of families in poverty, which has increased, as well as providing support to vulnerable groups that make a great contribution. Nonetheless, the out-of-pocket expenses for those who receive healthcare services are always estimated according to the demographic characteristics of the population, where 68.3% belong to the female gender and 31.7% belong to the male gender [34]. In addition, SIS affiliation helps to guarantee less out-of-pocket spending for poor rural populations, especially those that are vulnerable, and specific policies are needed to protect older adult populations in conditions of poverty, since in the health sector, out-of-pocket spending is capable of producing financial difficulties. These include catastrophic spending (own out-of-pocket expenses that usually exceed a household’s ability to pay) and impoverished health expenses (medical expenses are reduced below the indicated level) [33,35].

A particular situation was found for the specific case of the Puno region, where, through data collected from the ENAHO of the INEI for 2017, it was possible to determine that the heads of households incurred health expenses and that these expenses exceeded 20% of their income. Meanwhile, if the head of the household was affiliated with SIS but lived in a rural area and had a low educational level, the probability of repeating out-of-pocket spending was 28.49%, which was not the case if they were insured through ESSALUD [39,40,41,42].

Therefore, the research question is as follows: What were the main determinants of out-of-pocket spending on health in households in Peru in the times of the pandemic? In addition, the objective of this research was to establish the main determinants of out-of-pocket spending on health in households in Peru in the times of the pandemic. The research hypothesis is as follows: The main determinants of out-of-pocket spending on health in households in Peru in the times of the pandemic are the type of insurance, suffering from a disease, COVID-19 test results, health spending, permanent limitations, potable water service, presence of an older adult in the household, educational level, marital status of the head of the household, and area of residence.

## 2. Materials and Methods

### 2.1. Research Approach and Type

This scientific research uses a quantitative approach, since the use of information corresponds to variables and their quantifiable indicators. This type of research is nonexperimental, that is, there are no variations in or manipulations of the variables; only the phenomena are observed and analyzed in their natural environment. In addition, this research is cross-sectional, collecting data at a single moment to analyze and/or describe the variables [43,44].

### 2.2. Research Design

The research design is descriptive and correlational, since it relates two or more variables at a given moment. Correlation–causation designs measure the degree of relationship between two variables and do not manipulate the dependent variable, and their purpose is to describe the causes that may be generating a problematic situation [44,45].

### 2.3. Econometric Model

For the research hypothesis test, the use of the binomial logit econometric model was considered, where the dependent variable was out-of-pocket expenses, categorized as 1 if the household incurred out-of-pocket health expenses and 0 if the family did not incur any type of out-of-pocket health expense. The independent variables were as follows: the type of insurance that the household members had; whether a household member suffered from a disease; the results obtained from COVID-19 tests; whether the household incurred health expenses; whether any member of the household had permanent limitations; household drinking water service; the presence of an elderly person in the household; the educational level of the head of the household; the marital status of the head of the household; and the area of residence of the household, which can be rural or urban. For these variables, the following general econometric model was considered:$$\begin{array}{ll} P(Out-of- & pocket\;expense)\\ & =\beta \_0+\beta \_1\;Type\;of\;insurance+\beta \_2\;Suffering\;from\;some\;disease\\ & +\beta \_3\;Covid-19\;test+\beta \_4\;Health\;spending\\ & +\beta \_5\;Permanent\;limitations+\beta \_6\;Potable\;water\;service\\ & +\beta \_7\;Presence\;of\;older\;adult\;in\;the\;household+\beta \_8\;Education\;level\\ & +\beta \_9\;Marital\;status\;of\;the\;head\;of\;household+\beta \_10\;Residence\;area\\ & +U \end{array}$$

In this sense, when considering the binomial logit econometric model mathematically, since the dependent variable is binary, the corresponding representation is:P(Out−of−pocketexpense)=1/1+e^−(β_0+β_1 Type of insurance+β_2 Suffering from some disease+β_3 Covid−19 test+β_4 Health spending+β_5 Permanent limitations+β_6 Potable water service+β_7 Presence of older adult in the household+β_8 Education level+β_9 Marital status of the head of household+β_10 Residence area)

The details of the characteristics of the variables and their attributes are presented below (Table 1).

### 2.4. Data Collection Techniques and Instruments

This scientific research worked with information from a secondary source (existing information). Therefore, the technique used for data collection was the systematization of information from the National Household Survey (ENAHO) database of the National Institute of Statistics and Informatics (INEI) at the level of Peru for 2021. The statistical packages SPSS 25.0 and Stata 16.0 were used for information management and processing, and for obtaining the results.

In addition, since we worked with information from a secondary source, the data collection instruments were provided and validated by the INEI; therefore, it was not necessary to obtain an ethical approval document or ethical consideration for this study.

### 2.5. Population

The population studied was made up of the total number of respondents in the National Household Survey (ENAHO) conducted by the National Institute of Statistics and Informatics (INEI), which is applied annually throughout Peru, consolidating the entire population under study at the country level.

### 2.6. Type of Sample and Sample Obtained

Considering the information from the INEI, the sample considered corresponded to the existing data from the ENAHO-2021 database, where the probabilistic, area, stratified, multistage, and independent sampling designs were applied in each region of Peru, with a confidence level of 95% for the sample results.

## 3. Results

In general, the demographic characteristics of the population under study showed considerable behavior in the times of the pandemic. Of the total number of respondents who were considered in the present investigation, which amounts to 3493 respondents, 31.72% of the heads of households were female and 68.28% were male. In the case of marital status, 31.43% were cohabiting, 37.10% were married, 8.42% were widowed, 1.37% were divorced, 15.72% were separated; and only 5.95% were single. In the case of the age of the heads of households, the average was 50 years old, where the youngest head of a household was 19 years old and the oldest was 98 years old. In addition, in the case of the respondents’ level of education, 2.49% did not have any level of education, 64.01% had a primary or secondary level of education, 30.26% had a higher or university level of education, and only 3.24% undertook postgraduate studies. In the case of the area of residence, 19.10% of those surveyed resided in a rural area, and 80.90% resided in an urban area.

According to the processed results, households in Peru distributed their economic income to cover expenses on food, clothing, education services, household services, and health services, among others. Unlike other years, out-of-pocket spending in the times of COVID-19 was shared, but at the same time, in many cases, it was mainly faced by the heads of households, given that this was part of the unforeseen expenses and that it significantly affected the household budget. When showing this phenomenon, it is necessary to understand the behavior of the determining variables of out-of-pocket spending. An abstract of the indicators of these variables is detailed in Table 2.

The characteristics of the type of insurance contributed to out-of-pocket spending, since, on average, 48.64% of the population studied was insured through Social Health Security (ESSALUD), and 51.36% was insured through Comprehensive Health Insurance (SIS). Additionally, when evaluating said behavior and relationship, said variables showed a weak positive relationship, since Pearson’s ρ value was equal to 0.3048. In addition, when analyzing the relationship between out-of-pocket expenses and the type of insurance that the head of the household had, 62.02% of those who did not incur out-of-pocket expenses were insured through ESSALUD, while the remaining 37.98% were insured through SIS; on the other hand, 31.30% of those who did incur out-of-pocket expenses were insured through ESSALUD, while the remaining 68.70% were insured through SIS (Table 2, Table 3 and Table 12).

In the case of the relationship between out-of-pocket spending and whether a member of the household suffered from a disease, it was weak and positive, given that the value of Pearson’s ρ was equal to 0.1141; in addition, 51.76% of households had at least one member who suffered from a disease in the times of COVID-19. Complementarily, the aforementioned finding corroborates the relationship between these variables, since of the households that did not incur out-of-pocket expenses, 53.25% did not have a member suffering from a disease, while 46.75% did have a member suffering from a disease; on the other hand, of the households that did incur out-of-pocket expenses, 41.75% did not have a member suffering from a disease, while 58.25% did have a member suffering from a disease (Table 2, Table 4 and Table 12).

The relationship between out-of-pocket spending and positive COVID-19 test results for any member of the household was very weak, given that Pearson’s ρ value was equal to 0.0352, since 98.63% of those surveyed had negative COVID-19 test results. Additionally, of the households that did not incur out-of-pocket expenses, 98.99% had negative COVID-19 test results, and 1.01% had not yet received their results; on the other hand, of the households that did incur out-of-pocket expenses, 98.16% had negative COVID-19 test results, and 1.84% had not yet received their results (Table 2, Table 5 and Table 12).

Regarding the relationship between out-of-pocket spending and the existence of permanent limitations among the household members, it was very weak and positive, given that the value of Pearson’s ρ was equal to 0.1280. This is confirmed by the results in Table 2, where, on average, it is shown that 95.02% of the respondents did not suffer from any permanent limitation. In addition, of the households that did not incur out-of-pocket expenses, 97.46% had no limitations among their members, while 2.54% had a member with at least one permanent limitation; on the other hand, of the households that did incur out-of-pocket expenses, 91.85% had no limitations among their members, while 8.15% had a member with at least one permanent limitation (Table 6 and Table 12).

When looking at the relationship between out-of-pocket spending and the existence of a potable water service in the household, it was negative and weak, given that the value of Pearson’s ρ was equal to −0.1068. This is confirmed by the results in Table 2, since, on average, it is shown that 72.06% of the respondents had a potable water service in their household. The aforementioned finding is complemented by the relationship between these variables. Of the households that did not incur out-of-pocket expenses, 23.73% did not have a potable water service, while 76.27% did have a potable water service; on the other hand, of the households that did incur out-of-pocket expenses, 33.40% did not have a potable water service, while 66.60% did have a potable water service (Table 7 and Table 12).

Regarding the relationship between out-of-pocket spending and the presence of an older adult in the household, it was positive and weak, given that Pearson’s ρ value was equal to 0.1551. This is confirmed in Table 2, where, on average, it is shown that 26.28% of the households surveyed had at least one older adult in the household. Of the households that did not incur out-of-pocket expenses, 79.72% did not have at least one older adult, while 20.28% had at least one older adult; on the other hand, of the households that did incur out-of-pocket expenses, 65.94% did not have at least one older adult, while 34.06% had at least one older adult (Table 8 and Table 12).

In the case of the relationship between out-of-pocket spending and the educational level of the head of the household, it was negative and weak, since the value of Pearson’s ρ was equal to −0.1613. This is confirmed in Table 2, where, on average, it is shown that 65% of those surveyed had a primary or secondary level of education. Of the respondents with no educational level, 27.59% did not incur out-of-pocket expenses and 72.41% did incur out-of-pocket expenses; of the respondents with a primary or secondary educational level, 52.28% did not incur out-of-pocket expenses and 47.72% did incur out-of-pocket expenses; of the respondents with a higher or university level of education, 65.85% did not incur out-of-pocket expenses and 34.15% did incur out-of-pocket expenses; and of the respondents with a postgraduate educational level, 73.45% did not incur out-of-pocket expenses and 26.55% did incur out-of-pocket expenses (Table 9 and Table 12).

The relationship between out-of-pocket spending and the marital status of the head of the household was positive and very weak, since Pearson’s ρ value was equal to 0.0259. This is confirmed in Table 2, where, on average, it is shown that 37.10% of the respondents were married. Of the cohabiting respondents, 57.19% did not incur out-of-pocket expenses and 42.81% did incur out-of-pocket expenses; of the married respondents, 59.49% did not incur out-of-pocket expenses and 40.51% did incur out-of-pocket expenses; of the surveyed widows, 43.54% did not incur out-of-pocket expenses and 56.46% did incur out-of-pocket expenses; of the divorced respondents, 52.08% did not incur out-of-pocket expenses and 47.92% did incur out-of-pocket expenses; of the separated respondents, 55.01% did not incur out-of-pocket expenses and 44.99% did incur out-of-pocket expenses; and of the single respondents, 56.73% did not incur out-of-pocket expenses and 43.27% did incur out-of-pocket expenses (Table 10 and Table 12).

In the case of out-of-pocket spending and the area of residence of the household, the relationship was negative and weak, given that the value of Pearson’s ρ was equal to −0.1786. This is confirmed in Table 2, where, on average, it is shown that 80.90% of the households were located in urban areas, with the rest located in rural areas. In addition, of the households that did not incur in out-of-pocket expenses, 12.93% were located in rural areas, while 87.07% were located in urban areas; on the other hand, of the households that did incur out-of-pocket expenses, 27.09% were located in rural areas, while 72.91% were located in urban areas (Table 11 and Table 12).

In this sense, everything analyzed in Table 2, Table 3, Table 4, Table 5, Table 6, Table 7, Table 8, Table 9, Table 10 and Table 11 is abstracted below, demonstrating the existence of a consistent correlation between out-of-pocket spending and its determinants (Table 12).

**Table 12 ijerph-20-06759-t012:** Correlations between out-of-pocket spending and its determinants.

Variables	Out-of-Pocket Expenses	Type of Insurance	Suffering from a Disease	COVID-19 Test Results	Health Spending	Permanent Limitations	Potable Water Service	Presence of an Older Adult in the Household	Education Level	Marital Status of Head of Household	Residence Area
Out-of-pocket expenses	1.0000										
Type of insurance	0.3048	1.0000									
Suffering from a disease	0.1141	−0.0408	1.0000								
COVID-19 test results	0.0352	0.0361	0.0155	1.0000							
Health spending	0.2199	−0.1860	0.0911	−0.0315	1.0000						
Permanent limitations	0.1280	0.0359	0.1763	−0.0157	−0.0084	1.0000					
Potable water service	−0.1068	−0.2320	0.0449	−0.0525	0.1433	−0.0275	1.0000				
Presence of an older adult in the household	0.1551	−0.0436	0.1807	−0.0034	0.0741	0.1622	0.0616	1.0000			
Education level	−0.1613	−0.3609	0.0481	−0.0229	0.1828	−0.0871	0.1874	−0.1387	1.0000		
Marital status of head of household	0.0259	−0.0306	0.0781	0.0164	−0.0211	0.0089	0.1402	0.0218	0.0766	1.0000	
Residence area	−0.1786	−0.2659	0.0499	−0.0428	0.1334	−0.0327	0.4555	0.0170	0.2517	0.1584	1.0000

From analyzing the behaviors of the type of insurance, suffering from a disease, COVID-19 test results, health spending, permanent limitations among household members, potable water service, presence of an older adult in the household, educational level, marital status of the head of the household, area of residence, and the relationships they have with out-of-pocket spending, the results obtained after applying the binomial logit econometric model and the marginal effects of out-of-pocket spending are shown below (Table 13).

When analyzing the results of the statistics of the determinants of out-of-pocket spending in households in Peru in the times of COVID-19, they show adequate consistency. In the case of LR chi2(10), which is equal to 945.86, it shows that the coefficients of the type of insurance, suffering from a disease, COVID-19 test results, health spending, permanent limitations, potable water service, presence of an older adult in the household, educational level, marital status of the head of the household, and area of residence are jointly significant in explaining the probability of an increase or decrease in out-of-pocket spending in households in Peru (Table 13).

In addition, the pseudo R2 value of 0.1977 shows that 19.77% of the variation in out-of-pocket spending is explained by its aforementioned determinants. Complementarily, according to the statistics obtained from Estat class (correctly classified), the values are correctly classified, so it can be deduced that the model predicts 70.77% of the observations correctly; when considering the value of Fitstat (count R2), it is concluded that the fit of the model reaches 70.80% (Table 13).

In this sense, the type of insurance, suffering from a disease, COVID-19 test results, health spending, the existence of permanent limitations, the presence of an older adult in the household, and the marital status of the head of the household have a positive or direct relationship with out-of-pocket spending, which shows that, with an increase in the independent variables, there is a probability that out-of-pocket spending will increase in households in Peru. On the contrary, having a potable water service, educational level, and area of residence have a negative or indirect relationship with out-of-pocket expenses, thus showing that in the face of an increase in the independent variables, out-of-pocket spending will likely decrease. According to the marginal effects obtained from the binomial logit model (marginal effects after logit = 0.4299), the probability of out-of-pocket spending by households in Peru in the times of the pandemic is, on average, 42.99% (Table 13).

According to the marginal effects of the variables under analysis, we can conclude that if the type of insurance that the household members have is Comprehensive Health Insurance (SIS), then the probability that out-of-pocket spending will increase is 36.85 percentage points; if any member of the household has a disease, then the probability that out-of-pocket spending will increase is 8.48 percentage points; if the COVID-19 test result is negative, then the probability of increased out-of-pocket spending is 6.50 percentage points; if health spending is increased by PEN 1, then the probability that out-of-pocket spending will increase is 0.0065 percentage points; and if any member of the household has at least one permanent limitation, then the probability that out-of-pocket spending will increase is 23.73 percentage points (Table 13).

In the event that the household has a potable water service, then the probability that it will incur out-of-pocket expenses decreases by 4.81 percentage points; if there is at least one older adult in the household, then the probability that out-of-pocket spending will increase is 16.79 percentage points; if the educational level of the head of the household increases by one level, then the probability that they will incur out-of-pocket expenses decreases by 6.75 percentage points; if the marital status of the head of the household changes, then the probability that out-of-pocket spending will increase is 2.44 percentage points; and, finally, if the area of residence of the household is urban, then the probability of incurring out-of-pocket expenses decreases by 19.26 percentage points (Table 13).

## 4. Discussion

The results obtained in the present investigation are very similar to those obtained in the investigations carried out by Petrera et al. [33], given that these authors determined that only 5% of the population living in rural areas was treated in a hospital compared to 16% of the population living in urban areas. In addition, as in our research, out-of-pocket spending was directly related to the group of households that were affiliated with SIS, to health spending, and to the existence of at least one older adult in the household.

Our scientific research does not coincide with the results obtained by Torres and Knaul [46], since when they analyzed the variable of access to Comprehensive Health Insurance, it was a determinant but inversely related, while in our case it was positively related, to out-of-pocket expenses. However, our results do coincide when considering the variable of the existence of boys and girls or older adults in the household. Likewise, our results coincide when considering the area of residence, as in both investigations, the existence of a positive relationship with out-of-pocket spending was identified.

As with the previous studies, the results of the present investigation are closely related to the results obtained by Alvis-Zakzuk et al. [29] and Gil et al. [47], given that in said investigations, the determinants of out-of-pocket spending were the age of the head of the household, the existence of a permanent illness in a member of the household, the existence of a chronic disease in at least one member of the household, and the existence of an older adult in the household, an issue that was also determined in our scientific research.

Additionally, according to Pavón-León et al. [31], concerning the out-of-pocket expenses among older adults (60 years and over) who were insured through public insurance in Mexico, it was determined that monthly out-of-pocket spending on health rose to an average of USD 64.8, where the items where the highest expenses were demanded were medicines, coinciding with what was found in the present investigation. This also shows that despite the fact that household members have health insurance, there is inequity in access, mainly for the most vulnerable groups of households, which come mainly from rural areas, as demonstrated by Weid et al. [48] and Manzo and Talaga [49] in their scientific investigations.

Finally, the results found coincide, in some way, with what was determined by Jaafar et al. [50], Cutler [51], El-Khatib et al. [52], Hafidz et al. [53], and Chua et al. [54], where, in the times of COVID-19, they determined that out-of-pocket spending on health increased considerably above 30% on average, which was mainly destined for the treatment of the contagion caused by COVID-19; the same was found for other infectious diseases such as HIV, malaria, and tuberculosis, exposing the existing reality in many countries where there are greater social inequalities, as well as the growing gap between the rich and the poor, generating social, economic, and health crises, with a greater intensity in developing countries.

## 5. Conclusions

With the results obtained in the present investigation, it was possible to determine that the type of insurance, suffering from a disease, COVID-19 test results, health spending, the existence of permanent limitations, the presence of an elderly person in the household, and the marital status of the head of the household are determining factors and have a positive or direct influence on out-of-pocket spending on health in households in Peru; however, the drinking water service in households, the educational level of the head of the household, and area of residence are also determinants but have a negative or inverse influence on out-of-pocket spending on health in households in Peru.

Additionally, in households in Peru, the type of insurance determines out-of-pocket spending with 36.85 percentage units; suffering from a disease determines out-of-pocket spending with 8.48 percentage units; COVID-19 test results determine out-of-pocket spending with 6.50 percentage units; health spending determines out-of-pocket spending with 0.0065 percentage units; the existence of permanent limitations determines out-of-pocket spending with 23.73 percentage units; the existence of a potable water service in the household determines out-of-pocket spending with −4.81 percentage units; the presence of an older adult in the household determines out-of-pocket spending with 16.79 percentage units; educational level determines out-of-pocket spending with −6.75 percentage units; the marital status of the head of the household determines out-of-pocket spending with 2.44 percentage units; and the area of residence determines out-of-pocket spending with −19.26 percentage units.

In addition, as part of the present investigation, and according to the established limitations, it is recommended that policymakers take into account the issue of health, mainly in households with low economic resources, since the greater expenditure destined for actions or unforeseen activities makes it difficult to cover the basic food basket in the household, which compromises the quality-of-life conditions of the household members. It is also recommended that scientific researchers continue to deepen studies with this approach and characteristics, considering new topics that contribute to the health sector, since from these topics public policies can be designed for the benefit of society.

## Figures and Tables

**Table 1 ijerph-20-06759-t001:** Description of the variables identified in this scientific research.

Variable	Variable Type	Categorization
Out-of-pocket expenses	Qualitative	1: Incurred out-of-pocket expenses 0: Did not incur out-of-pocket expenses
Type of insurance	Qualitative	1: Insured through SIS 0: Insured through ESSALUD
Suffering from a disease	Qualitative	1: Yes, suffered 0: Did not suffer
COVID-19 test results	Qualitative	1: Negative result 2: Positive result 3: Did not receive the results
Health spending	Quantitative	Numeric
Permanent limitations	Qualitative	1: Had at least one permanent limitation 0: Had no permanent limitations
Potable water service	Qualitative	1: Had potable water in the household 0: Did not have potable water in the household
Presence of older adult in the household	Qualitative	1: Had an elderly person in the household 0: Did not have an elderly person in the household
Education level	Qualitative	1: No educational level 2: Primary or secondary educational level 3: Higher or university educational level 4: Postgraduate level of studies
Marital status of head of household	Qualitative	1: Cohabitant 2: Married 3: Widow(er) 4: Divorced 5: Separated (a) 6: Single
Residence area	Qualitative	1: Rural area 2: Urban area

**Table 2 ijerph-20-06759-t002:** Description of the variables identified in this scientific research.

Variable	Mean	Standard Deviation	Minimum Value	Maximum Value	Asymmetry	Kurtosis
Out-of-pocket expenses	0.4354	0.4959	0	1	0.2604	1.0678
Type of insurance	0.5136	0.4999	0	1	−0.0544	1.0030
Suffering from a disease	0.5176	0.4998	0	1	−0.0705	1.0050
COVID-19 test results	1.0275	0.2329	1	3	8.3537	70.7848
Health spending	2400.7620	3861.7210	0	42,509.76	4.0197	26.9875
Permanent limitations	0.0498	0.2176	0	1	4.1385	18.1271
Potable water service	0.7206	0.4488	0	1	−0.9832	1.9667
Presence of an older adult in the household	0.2628	0.4402	0	1	1.0777	2.1615
Education level	2.3424	0.5829	1	4	0.7448	3.3844
Marital status of head of household	2.5070	1.6014	1	6	0.9512	2.5339
Residence area	0.8090	0.3931	0	1	−1.5725	3.4729

**Table 3 ijerph-20-06759-t003:** Characteristics of the type of insurance.

Variable		Type of Insurance		Total
	Category	Insured through ESSALUD	Insured through SIS	
Out-of-pocket expenses	Did not incur out-of-pocket expenses	1223	749	1972
	%	62.02	37.98	100
	Incurred out-of-pocket expenses	476	1045	1521
	%	31.3	68.7	100
Total		1699	1794	3493
		48.64	51.36	100

**Table 4 ijerph-20-06759-t004:** Household member suffering from a disease.

Variable		Suffering from a Disease		Total
	Category	No	Yes	
Out-of-pocket expenses	Did not incur out-of-pocket expenses	1050	922	1972
	%	53.25	46.75	100
	Incurred out-of-pocket expenses	635	886	1521
	%	41.75	58.25	100
Total		1685	1808	3493
		48.24	51.76	100

**Table 5 ijerph-20-06759-t005:** COVID-19 test results.

Variable		COVID-19 Test Results		Total
	Category	Negative	Did Not Have the Results	
Out-of-pocket expenses	Did not incur out-of-pocket expenses	1952	20	1972
	%	98.99	1.01	100
	Incurred out-of-pocket expenses	1493	28	1521
	%	98.16	1.84	100
Total		3445	48	3493
		98.63	1.37	100

**Table 6 ijerph-20-06759-t006:** Existence of permanent limitations among household members.

Variable		Permanent Limitations		Total
	Category	No Limitations	At Least One Permanent Limitation	
Out-of-pocket expenses	Did not incur out-of-pocket expenses	1922	50	1972
	%	97.46	2.54	100
	Incurred out-of-pocket expenses	1397	124	1521
	%	91.85	8.15	100
Total		3319	174	3493
		95.02	4.98	100

**Table 7 ijerph-20-06759-t007:** Potable water service in the household.

Variable		Potable Water Service		Total
	Category	Did Not Have Potable Water Service	Did Have Potable Water Service	
Out-of-pocket expenses	Did not incur out-of-pocket expenses	468	1504	1972
	%	23.73	76.27	100
	Incurred out-of-pocket expenses	508	1013	1521
	%	33.40	66.60	100
Total		976	2517	3493
		27.94	72.06	100

**Table 8 ijerph-20-06759-t008:** Presence of an older adult in the household.

Variable		Presence of an Older Adult in the Household		Total
	Category	No	Yes	
Out-of-pocket expenses	Did not incur out-of-pocket expenses	1572	400	1972
	%	79.72	20.28	100
	Incurred out-of-pocket expenses	1003	518	1521
	%	65.94	34.06	100
Total		2575	918	3493
		73.72	26.28	100

**Table 9 ijerph-20-06759-t009:** Educational level of the head of the household.

Variable		Out-of-Pocket Expenses		Total
	Category	Did Not Incur Out-of-Pocket Expenses	Incurred Out-of-Pocket Expenses	
Education level	No educational level	24	63	87
	%	27.59	72.41	100
	Primary or secondary education	1169	1067	2236
	%	52.28	47.72	100
	Higher or university education	696	361	1057
	%	65.85	34.15	100
	Postgraduate studies	83	30	113
	%	73.45	26.55	100
Total		1972	1521	3493
		56.46	43.54	100

**Table 10 ijerph-20-06759-t010:** Marital status of the head of the household.

Variable		Out-of-Pocket Expenses		Total
	Category	Did Not Incur Out-of-Pocket Expenses	Incurred Out-of-Pocket Expenses	
Marital status of the head of the household	Cohabitant	628	470	1098
	%	57.19	42.81	100
	Married	771	525	1296
	%	59.49	40.51	100
	Widow(er)	128	166	294
	%	43.54	56.46	100
	Divorced	25	23	48
	%	52.08	47.92	100
	Separated	302	247	549
	%	55.01	44.99	100
	Single	118	90	208
	%	56.73	43.27	100
Total		1972	1521	3493
		56.46	43.54	100

**Table 11 ijerph-20-06759-t011:** Area of residence of the household.

Variable		Residence Area		Total
	Category	Rural	Urban	
Out-of-pocket expenses	Did not incur out-of-pocket expenses	255	1717	1972
	%	12.93	87.07	100
	Incurred out-of-pocket expenses	412	1109	1521
	%	27.09	72.91	100
Total		667	2826	3493
		19.10	80.90	100

**Table 13 ijerph-20-06759-t013:** Regression results of the binomial logit model and marginal effects of out-of-pocket spending and its determinants.

Out-of-Pocket Expenses	Coefficients	Z-Value	Probability	Marginal Effects
Type of insurance	1.5809	19.33	0.00	0.368523
Suffering from a disease	0.3475	4.29	0.00	0.084898
COVID-19 test results	0.2653	1.63	0.10	0.065031
Health spending	0.0003	15.76	0.00	0.000065
Permanent limitations	0.9736	5.31	0.00	0.237341
Potable water service	−0.1954	−1.98	0.05	−0.048129
Presence of an older adult in the household	0.6805	7.35	0.00	0.167921
Education level	−0.2755	−3.52	0.00	−0.067520
Marital status of head of household	0.0997	4.01	0.00	0.024432
Residence area	−0.7806	−7.11	0.00	−0.192640
Constant	−1.2403	−4.46	0.00	
Number of observations	3493	Pseudo R2		0.1977
LR chi2(10)	945.86	Log likelihood		−1919.0381
Prob > chi2	0.00000	Marginal effects after logit		0.4299
Estat class (correctly classified)	70.77%	Fitstat (count R2)		0.708

## Data Availability

The data used to prepare this scientific article are available in the database of the National Household Survey (ENAHO) of the National Institute of Statistics and Informatics (INEI), Peru.

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
