# Peer review of "Determinants of Out-of-Pocket Health Spending in Households in Peru in the Times of the Pandemic (COVID-19)"

_ijerph, 2023, doi:10.3390/ijerph20186759_

Round 1
Reviewer 1 Report
Determinants of out-of-pocket health spending in households in Peru in times of pandemic (Covid-19)
Peer Reviewer – Comments
Dear Authors,
Here are my few comments to address. They are as follows:
1. From the topic, define household in the introduction section because household varies and differs across different geographical settings. For instance, In 2021, South Africa had a population of approximately 60,5 million. Although some South Africans resided in institutions such as hospitals and old age homes, SA had an estimated 18 million households, with an average household size of 3,34 persons. Also, there are types of households – family households and non-family households. Are you referring to family households or both (family and non-family households)?
Also, clarify and conceptualize out-of-pocket spending. See these Readings below:
Koch SF, Setshegetso N (2020) Catastrophic health expenditures arising from out-of-pocket payments: Evidence from South African income and expenditure surveys. PLoS ONE 15(8): e0237217. https://doi.org/10.1371/journal.pone.0237217.
Sirag, A., & Mohamed Nor, N. (2021). Out-of-Pocket Health Expenditure and Poverty: Evidence from a Dynamic Panel Threshold Analysis. Healthcare (Basel, Switzerland), 9(5), 536. https://doi.org/10.3390/healthcare9050536
OECD (2009), “Burden of out-of-pocket health expenditure”, in Health at a Glance 2009: OECD Indicators, OECD Publishing, Paris. DOI: https://doi.org/10.1000787/health_glance-2009-62-en.
Tur-Sinai, A. (2022). Out-of-Pocket Expenditure on Medical Services Among Older Adults: A Longitudinal Analysis. Frontiers in Public Health, 10, 836675. https://doi.org/10.3389/fpub h.2022.836675.
Jalali, F.S., Bikineh, P. & Delavari, S. Strategies for reducing out of pocket payments in the health system: a scoping review. Cost Eff Resour Alloc 19, 47 (2021). https://doi.org/10.1186/s12962-021-00301-8.
2. The sentence in the abstract is too long. Can you break the sentence into two to make its meaning more straightforward [Line 22-24]
3. Give the whole meaning of this word ESSALUD [Line 61]
4. The sentence is too long; kindly revisit it [Line 67-76].
5. You only mentioned the United States of America regarding out-of-payment from pockets for medical expenses. Why did you not include studies from African countries? Your systematic literature review should follow a ‘funnel approach’ from developed to developing countries, then to Peru. This will help in identifying the reasons and different factors associated with out-of-pocket health spending in households in developed countries and developing countries as well as in Peru because you will see the differences [Line 48-154]
6. What do you mean by the scientific research type and approach? What are you trying to explain here [157-162]? There are different research designs, and you must understand what research design all is about. Take a look at this below:
What is research design? Research design is the method that a researcher selects to organize their research project or study. Research designs can provide instructions for collecting, analyzing and measuring data effectively. Using a research design is essential because: it can help you ensure that your research addresses your research problem; it acts as an outline and guide for the entire research project, and it can help you organize all the different components of your research project. You can choose an effective research design by considering your research problem, topic, or knowledge gap your research aims to address. Usually, researchers include their research question and design selection in the introduction of their research paper.
Types of research design: You can choose a research design by reviewing other research papers' methods and learning about different research designs. Here are 20 types of research design that you can consider using for your research project:
1. Exploratory research design: One common type of research design is exploratory design. The exploratory research design format is proper when you don't have a clearly defined problem. This type of research design is often less structured than other research design options, and you can use it as a guide for your initial research to uncover your research problem.
2. Observational research design: Observational design is also a common type of research design. The observational research design format emphasizes observing your research topic without altering variables. Using an observational research design, you can observe and record behaviours or phenomena rather than experiment.
3. Descriptive research design: Descriptive design is another type of research design. The goal of using a descriptive research design is to describe a research topic, so this type of research is proper when you need more information about your topic. Descriptive research design can also help you understand the "what," "where," "when", and "how" of your research topic. The one question that a descriptive research design does not answer is "why."
4. Case study: Another type of observational research design is the case study format. Case studies analyse real-world situations to understand and evaluate past problems and solutions. Therefore, case studies are helpful when you want to test how an idea applies to real life, and this research design is trendy in marketing, advertising and social science. The five-part case study format includes: Title, Overview, Problem, Solution and Results
5. Action research design: Another type of research design is the action research design. The action research design format involves initial exploratory analysis and the development of an action strategy. This design format is collaborative and focuses on finding solutions, making it practical for many research topics. You can use the action research design to solve real problems.
6. Experimental research design: Experimental research design is also standard. The experimental research design is advantageous when testing how different factors affect a situation, making this design type very versatile. The experimental research design uses the scientific method, which includes elements like Hypothesis: A research hypothesis is a statement that describes what you predict your research to reveal; Independent variable: An independent variable is a variable that does not depend on other variables; Dependent variable: A dependent variable is a variable that depends on another variable; Control variable: A control variable is a variable that remains constant throughout a research experiment.
7. Causal research design: The causal research design is another research design that researchers commonly choose. The causal research design format attempts to identify and understand relationships between variables, which can be valuable across many industries. Causal research designs typically involve at least two variables and explore many possible reasons for a relationship between variables.
8. Correlational research design: The correlational research design is also commonly used along with the causal research design. The correlational research design format identifies relationships between variables like the causal format. When you use a correlational research design, you measure variables but do not alter them.
9. Diagnostic research design: Another type of research design is the diagnostic research design. The diagnostic research design attempts to find the underlying factors that cause events or phenomena. This research type is helpful to help you understand what's causing problems so you can find solutions.
10. Cross-sectional research design: Cross-sectional design is another type of observational research design. The cross-sectional research design involves observing multiple individuals at the same point in time. This research type does not alter variables.
11. Sequential research design: Sequential research design is another helpful type of research design. The sequential research design format divides research into stages, and each stage builds on the last. Therefore, you can complete sequential research at multiple points in time, allowing you to study phenomena that occur over periods.
12. Cohort research design: Cohort research design, a type of observational research, is another research design type. This type of research design is commonly used in medicine, but it can also have applications in other industries. Cohort design involves examining subjects who have already been exposed to a research topic, making it especially effective for conducting ethical research on medical topics or risk factors. This design type is very flexible and applies to both primary and secondary data.
13. Historical research design: Researchers can also use historical research design. The historical research design allows you to use past data to test your hypothesis. Historical research relies on historical data like archives, maps, diaries and logs. This research design can be beneficial for completing trend analysis or gathering context for a research problem.
14. Field research design: Another type of research design is the field research design. The field research design, a qualitative research method, allows you to observe subjects in natural environments. This can allow you to collect data directly from real-world situations.
15. Systematic review: Systematic review is another type of research design. Completing a systematic review involves reviewing evidence and analyzing data from existing studies. This can allow you to use previous research to come up with new conclusions.
16. Survey: Researchers also use the survey research design frequently. You can use surveys to gather information directly from your sample population. Some types of surveys include Interviews: Interviews are one popular type of survey. Interviews allow you to ask questions to a research subject one-on-one, which can allow you to ask follow-up questions and gain additional insights; Online forms: You can also use online forms to conduct surveys. You can use many websites or software programs to create intuitive online forms with various question types, including short-answer and multiple-choice; Focus groups: Focus groups are another critical survey method. Using focus groups, you can facilitate discussions with a group of research subjects to gain valuable research insights from your sample population; Questionnaires: Another type of survey is a questionnaire. In a questionnaire, you can list questions for a research subject to answer, making this an effective data collection method.
17. Meta-analysis research design: Meta-analysis is a quantitative research design. The meta-analysis research design format uses a variety of populations from different existing studies. This means this method allows you to use previous research to form new conclusions.
18. Mixed-method research design: Researchers can also use a mixed-method research design. Mixed-method research designs combine multiple methods to create the best path for a specific research project. This type of research can include both qualitative and quantitative research methods.
19. Longitudinal research design: Another quantitative, observational research design type is longitudinal design. The longitudinal research design involves observing the same sample repeatedly over some time. Depending on your research, this period might be anywhere from a few weeks to several decades.
20. Philosophical research design: Philosophical research design is another research design type. The philosophical research design can help you analyze and understand your research problem. This design type builds on philosophical argumentation techniques. The three key areas of philosophical research design are Epistemology: Epistemology focuses on knowledge and certainty; Ontology: Ontology focuses on human nature and existence; and Axiology: Axiology is the study of values, which applies primarily to ethics. The philosophical research design can help you understand research purposes, make ethical decisions, and think critically about your research topic.
So, you have to carefully pick which one or two of the research designs are suitable for your study.
7. The Econometric model should be under variable measurements where you discuss the dependent and independent variables. The Econometric Model are always the dependent variable [Line 169-183]
8. Explain in detail the data collection techniques and instruments [Line 184-186]
9. Who are the population? Can you discuss this session in detail? [Line 187-189]
10. What do you mean by the type of sample and sample obtained? Can you clarify this in detail? [Line 190-193]
11. What is your outcome variable(s) and the explanatory variables. Kindly state them clearly and explain them in sequence
12. Explain in-depth detail the method of data collection.
13. Where is the ethical approval or Ethical consideration for this study?
14. The Method section should follow these steps as follows: Study design, Study setting, Population sampling, Sample size, Sampling strategy including inclusion and exclusion criteria, Variable measurement(s), Data collection, Ethical consideration and Data analysis
15. What type of analyses did you utilize in this study? Please, provide a section for it under Method Section.
16. In the Result Section, insert a Table of demographic characteristics for the study respondents. [Line 195]
17. Run and Align your analysis with your study objectives. Let the objectives follow in sequence to show the findings from the results. [Line 195-385]
18. Align the Discussion with the study finding. Let the discussion stem from the study results by objectives. The objectives will make the paragraphs, which must be brief and straight to the point. [Line 385-404]
19. Where is the section for the Strengths and Limitations of the Study? [Line 405]
20. Where are the recommendations for this study? [Line 405]
21. Rewrite the conclusions to reflect the findings of your study. [Line 406-420]
22. Kindly submit this manuscript to a Professional English Editor to edit the entire manuscript.
Submit this manuscript to a Professional English Editor to edit the entire manuscript for better clarity and to reduce the long sentences.
Author Response
1. Building on the topic, define household in the introduction section because household varies and differs in different geographic settings. For example, in 2021, South Africa had a population of approximately 60.5 million. Although some South Africans resided in institutions such as hospitals and nursing homes, SA had approximately 18 million households, with an average household size of 3.34 people. In addition, there are types of households: family households and non-family households. Are you referring to family households or both (family and non-family households)?
They are family homes
Also, clarify and conceptualize out-of-pocket spending. See these readings below:
According to the suggestion, the list of recommended articles was considered and cited in this scientific article.
2. The abstract sentence is too long. Can you split the sentence in two to make its meaning clearer? [Lines 22-24]
The corresponding adjustment was made according to the guidelines of scientific articles.
3. Give the full meaning of this word ESSALUD [Line 61]
What was requested was done.
4. The sentence is too long; kindly revisit it [Lines 67-76].
Corrected what was indicated.
5. You only mentioned the United States of America in terms of out-of-pocket payment for medical expenses. Why didn't you include studies from African countries? Your systematic review of the literature should follow a "funnel approach" from developed countries to developing countries, and then to Peru. This will help to identify the reasons and the different factors associated with out-of-pocket spending on health in households in developed and developing countries, as well as in Peru because they will see the differences [Lines 48-154]
Expanded what was indicated for further references.
6. What do you understand by type and approach of scientific research? What are you trying to explain here [157-162]? There are different research designs, and you need to understand what the research design is all about. Take a look at this below:
It corresponds to a quantitative approach, with a descriptive and correlational research design.
7. The econometric model must be under variable measures where the dependent and independent variables are analyzed. The Econometric Model is always the dependent variable [Lines 169-183]
The detail of the Logit-Binomial econometric model was specified.
8. Explain in detail the data collection techniques and instruments [Lines 184-186]
What was requested was specified.
9. Who are the population? Can you discuss this session in detail? [Line 187-189]
The requested is specified.
10. What do you understand by type of sample and sample obtained? Can you clarify this in detail? [Line 190-193]
It was corrected, there was a typing error, it was corrected with the sample design.
11. What are your outcome variable(s) and explanatory variables? Kindly write them clearly and explain them in sequence.
The specification of the variables is presented in Table 1.
12. Explain in detail the data collection method.
The same is specified.
13. Where is the ethical approval or ethical consideration for this study?
In this case it does not correspond and was justified in the investigation.
14. The Method section should follow these steps as follows: study design, study setting, population sampling, sample size, sampling strategy, including inclusion and exclusion criteria, variable measurement(s) , data collection, ethical consideration and data analysis .
Recommended was considered.
15. What type of analysis did you use in this study? Provide a section for it in the Method Section.
The same was specified.
16. In the results section, insert a table of demographic characteristics for the survey respondents. [Line 195]
An analysis of demographic characteristics was made.
17. Run and align your analysis with your study objectives. Let the objectives follow in sequence to show the findings of the results. [Line 195-385]
It is considered the same.
18. Align the discussion with the finding of the study. Let the discussion start from the results of the study by objectives. The objectives are made up of the paragraphs, which should be short and direct to the point. [Line 385-404]
This point was expanded and the recommendations were considered.
19. Where is the Strengths and Limitations of the Study section? [Line 405]
The requested is included.
20. Where are the recommendations for this study? [Line 405]
The recommendations were considered.
21. Rewrite the conclusions to reflect the findings of your study. [Line 406-420]
The conclusions were written in a better way.
22. Kindly send this manuscript to a professional English editor for full manuscript editing.
The indicated was considered.
For more details, the corrected version of the scientific paper is attached.

Reviewer 2 Report
The paper addresses an essential issue of out-of-pocket health expenses in Peruvian households, explicitly focusing on the Covid-19 pandemic. The research draws from a comprehensive dataset, employing a quantitative, cross-sectional approach and using a logit-binomial econometric model.
Strengths:
1. The use of the ENAHO 2021 database of the INEI lends credibility to the research, given its comprehensive and reputable nature.
2. The paper delves deep into the various determinants of out-of-pocket spending, providing percentages for each factor's influence.
3. The study is of high importance in the current global context, providing insights into the financial implications of the pandemic on households.
Drawbacks:
1. The paper solely concentrates on Peruvian households, which, while necessary, limits the generalizability of the findings.
2. The abstract, though comprehensive, can be overwhelming with its barrage of statistics and factors. A more precise and more concise presentation would be preferable.
3. While the paper employs a quantitative, cross-sectional approach, it would benefit from a more detailed elaboration on the methodology, especially the rationale behind selecting the logit-binomial econometric model.
4. The paper could benefit from more background information on Peruvian households, their general expenditure patterns, and the broader healthcare system.
5. There is no mention of other potential external factors or events that might have impacted the out-of-pocket expenses in 2021.
Recommendations:
1. Future research could include comparisons with other countries, providing a more comprehensive understanding. It is recommended to include other countries in the Current research analysis section.
2. More details and justifications should be provided regarding the chosen methodological design.
3. Along with the quantitative approach, integrating qualitative data through interviews or surveys might provide richer insights into the household's decision-making processes.
4. It is advisable to present the abstract more reader-friendly, perhaps by segmenting the factors and percentages or using visual aids.
5. The research should consider other significant events or external factors in 2021 that could have impacted household expenditures.
6. A section elaborating on Peru's socioeconomic and health infrastructure context would be helpful for readers unfamiliar with the region.
7. The paper should explore the factors that negatively determine out-of-pocket expenses, explaining the reasons and implications.
8. It is recommended to expand the Discussion section.
This paper provides valuable insights into the determinants of out-of-pocket health expenses in Peruvian households during the pandemic. However, there is room for improvement in its presentation, scope, and depth. The recommendations, if incorporated, would make the research more comprehensive, accessible, and valuable to a broader audience.
Author Response
1. The document focuses only on Peruvian households, which, while necessary, limits the generalizability of the findings.
What is indicated is correct, and it is considered in recommendations to expand other similar studies at the level of other countries.
2. The summary, while comprehensive, can be overwhelming with its barrage of statistics and factors. A more precise and concise presentation would be preferable.
The abstract was corrected, as indicated.
3. Although the paper employs a quantitative cross-sectional approach, it would benefit from further elaboration of the methodology, especially the rationale behind the selection of the logit-binomial econometric model.
The justification and use of the econometric model was detailed.
4. The document could benefit from more background information on Peruvian households, their general spending patterns, and the health system in general.
Further background was considered in this article.
5. There is no mention of other possible external factors or events that could have affected out-of-pocket expenses in 2021.
The approach to the identified problem was expanded with indicators and greater support.
Recommendations:
1. Future research could include comparisons with other countries, providing a fuller understanding. It is recommended to include other countries in the Current Research Analysis section.
It is correct and was included in the recommendations.
2. More details and justifications should be provided about the chosen methodological design.
The methodological design was better justified.
3. Along with the quantitative approach, the integration of qualitative data through interviews or surveys could provide more detailed information on household decision-making processes.
The source of information that is secondary was justified.
4. It is advisable to present the summary in an easier to read way, perhaps by segmenting the factors and percentages or using visual aids.
The abstract of the investigation was improved.
5. The investigation must consider other significant events or external factors in 2021 that could have affected household expenses.
what was indicated was taken into account.
6. A section explaining the socioeconomic and health infrastructure context of Peru would be useful for readers unfamiliar with the region.
A paragraph was incorporated in the first part of the results, as in the introduction.
7. The paper should explore the factors that negatively determine out-of-pocket expenses, explaining the reasons and implications.
It was taken into account.
8. It is recommended to expand the Discussion section.
This section was expanded and is as indicated.
For further details, the corrected version of the scientific article is attached.

Round 2
Reviewer 1 Report
None
Reviewer 2 Report
Thanks for the authors for considering reviewers' comments and recommendations.